# Identification and Characterization of Human Breast Milk and Infant Fecal Cultivable Lactobacilli Isolated in Bulgaria: A Pilot Study

**DOI:** 10.3390/microorganisms12091839

**Published:** 2024-09-05

**Authors:** Asya Asenova, Hristiyana Hristova, Stanimira Ivanova, Viliana Miteva, Ivelina Zhivkova, Katerina Stefanova, Penka Moncheva, Trayana Nedeva, Zoltan Urshev, Victoria Marinova-Yordanova, Tzveta Georgieva, Margarita Tzenova, Maria Russinova, Tzvetomira Borisova, Deyan Donchev, Petya Hristova, Iliyana Rasheva

**Affiliations:** 1Department of General and Industrial Microbiology, Faculty of Biology, Sofia University St. Kliment Ohridski, Dragan Tsankov Blvd 8, 1164 Sofia, Bulgaria; a.asenova_97@abv.bg (A.A.); nedeva@biofac.uni-sofia.bg (T.N.); p.hristova@biofac.uni-sofia.bg (P.H.); 2Department of Clinical Microbiology, National Center of Infectious and Parasitic Disease, Yanko Sakuzov Blvd 26, 1504 Sofia, Bulgariadeyandonchev@ncipd.org (D.D.); 3Agrobioinstitute Bulgarian Agriculture Academy, Dragan Tsankov Blvd 8, 1164 Sofia, Bulgaria; katerina_stefanova@abi.bg; 4LB Bulgaricum PLC, Malashevska Str. 14, 1113 Sofia, Bulgaria; 5Department of Bioactivity of Compouds, Centre of Competence “Sustainable Utilization of Bio Resources and Waste of Medicinal and Aromatic Plants for Innovative Bioactive Products”, Dragan Tsankov Blvd 8, 1164 Sofia, Bulgaria; 6Department of Applied Genomics and GMO, National Center of Public Health and Analyses, Academic Ivan Geshov Blvd 15, 1431 Sofia, Bulgaria; tzv.georgieva@ncpha.government.bg; 7Human Milk Bank, Sava Mihailov Str. 57, 1309 Sofia, Bulgariacv_borisova@abv.bg (T.B.); 8Centre of Competence “Fundamental Translational and Clinical Research in Infection and Immunity”, Yanko Sakuzov Blvd 26, 1504 Sofia, Bulgaria

**Keywords:** lactobacilli, MALDI-TOF MS, 16S rRNA, human breast milk, infant feces, RAPD-PCR DNA fingerprints

## Abstract

During the last few decades, the main focus of numerous studies has been on the human breast milk microbiota and its influence on the infant intestinal microbiota and overall health. The presence of lactic acid bacteria in breast milk affects both the quantitative and qualitative composition of the infant gut microbiota. The aim of this study was to assess the most frequently detected cultivable rod-shaped lactobacilli, specific for breast milk of healthy Bulgarian women and fecal samples of their infants over the first month of life, in 14 mother–infant tandem pairs. Additionally, we evaluated the strain diversity among the most common isolated species. A total of 68 Gram-positive and catalase-negative strains were subjected to identification using the MALDI-TOF technique. Predominant cultivable populations belonging to the rod-shaped lactic acid bacteria have been identified as *Lacticaseibacillus rhamnosus*, *Limosilactobacillus fermentum*, *Lacticaseibacillus paracasei*, and *Limosilactobacillus reuteri*. Also, we confirmed the presence of *Lactiplantibacillus plantarum* and *Lactobacillus gasseri.* Up to 26 isolates were selected as representatives and analyzed by 16S rRNA sequencing for strain identity confirmation and a phylogenetic tree based on 16S rRNA gene sequence was constructed. Comparative analysis by four RAPD primers revealed genetic differences between newly isolated predominant *L. rhamnosus* strains. This pilot study provides data for the current first report concerning the investigation of the characteristic cultivable lactobacilli isolated from human breast milk and infant feces in Bulgaria.

## 1. Introduction

Breast milk, considered the golden standard for infant nutrition, plays the vital role of modulating the gut microbiome of newborns in a health-determining way that influences their well-being early on and later in life. The presence of bioactive components and microorganisms in human breast milk distinguishes it from powdered formula. These components play a crucial role in developing the gut microbiome and immunity of the newborns. Therefore, any fluctuations that may arise, for instance, mixed-feeding, solely formula feeding, and antibiotic or probiotic consumption, contribute to positive or negative gut community development deviations from the “golden standard” feeding. The potentially probiotic microflora of breast milk manifests different characteristics depending on many factors, the majority of which related to the individuality of the donors, their lifestyle and geographical location.

Lactic acid bacteria (LAB) have accumulated scientific and commercial popularity during the last decades due to their beneficial presence in the microbiota of the human body, specifically the gastrointestinal tract (GIT). The metabolic advantages of many probiotic strains, along with their protective capabilities against infection agents, have been the focus of a number of scientific studies, especially when it comes to neonates.

The presence and persistence of these potentially beneficial bacteria in the infant’s gut depend on a number of factors, including the mother’s microbiota, mode of delivery, type of feeding, intake of probiotics or antibiotics, geographic, and other characteristics. The inoculation of the neonate begins before birth, as shown by studies proving the existence of bacteria in the amniotic fluid, placenta, and meconium [1]. The type of birth is considered crucial for the infant’s initial GIT colonization due to the different bacteria acquired during birth, characteristic of either the mother’s skin microbiota in a cesarean section (CS) or the vaginal microbiota during a spontaneous natural delivery. The dominance of *Bifidobacterium*, *Bacteroides*, and *Lactobacillaceae*, along with higher variability is well represented in vaginally born infants, whereas lower diversity, *Staphylococcus*, *Streptococcus* and *Clostridium* are attributed to a cesarean delivery [2]. Not only does the transfer of beneficial microorganisms during the vaginal birth differ from a CS delivery, but some immune-stimulatory factors, such as interleukin 18 (IL-18) and tumor necrosis factor (TNF-α), are also absent in the cesarean babies [3].

Breastfeeding influences the gut ecosystem tremendously by providing the neonate with the most appropriate selection of nutrients for the growing organism. Ma et al. [4] proved the beneficial microbiological qualities of the human milk compared to the infant formula and displayed the nutritional capacity of breast milk to not only deliver the needed elements but also to formulate a healthy GIT microbiological profile. The modulation of an infant immune system increasingly depends on the probiotic potential of the representatives in the gut. The commensal bacteria balance the Th1/Th2 response, enhance the anti-inflammatory response [5], affect the production of sIgA antibodies, and play a significant part in the development of the gut-associated lymphoid tissue [6]. Neonatal antibiotic prescription is one of the most prominent causes of dysbiosis [7], which is related to necrotizing enterocolitis, inflammatory diseases, and obesity [8]. Neonates who have been on antibiotics show lower amounts of commensal microflora, such as members of the genera *Bifidobacterium* and *Bacteroides* [9,10,11], *Firmicutes*, along with higher *Proteobacteria* [12] and *Enterobacteriaceae* [13], resulting in microbiological deficiency. On the other hand, infant probiotic intake predisposes positive alternations in the gut microbiota and immune responses [14], as well as mother’s probiotic supplementation [15]. Other circumstances, such as pre-term pregnancy and prolonged hospitalization induce variation in the gut microbiota orchestration in newborns. They are related to the use of antibiotics, physiological immaturity, and lower birth weight, along with the breastfeeding difficulty, which is characteristic of a pre-term pregnancy, and contribute to the *Enterobacteriaceae* and *Clostridium* domination of the GIT, which is related to intestine disorder development, for instance, necrotizing enterocolitis [6,16,17]. The enormous variety of factors that affect the infant gut inoculation and maturation assemble a complex system that requires further scientific research.

Since breast milk, the main nutritious source of neonates, was evidenced as a non-sterile fluid, the following two hypotheses on the origin of human milk microbiota have emerged: the hypothesis of the retrograde flow and the other of the entero-mammary pathway [7]. The retrograde flow represents the potential of bacterial transfer from the oral cavity of a breastfed neonate to the mammary duct system of the mother [18]. The bacteria entero-mammary route suggests that non-pathogenic bacterial cells penetrate dendritic cells and macrophages in the GIT of the mother, which subjects them to being transferred to the mammary gland via the lymph nodes [6]. Numerous studies have confirmed these theories; nonetheless, many individual factors significantly influence each case of mother–infant microbiota transmission [19,20,21].

Geographic location has been proven to be another determinant of dominating microbiota in newborns. Several regions have already published information regarding the distinctive characteristics of its population, especially with respect to probiotic microbiota [18,22,23,24,25,26,27]. Rod-shaped, Gram-positive lactobacilli were found on many occasions in human milk and infant fecal samples; moreover, the different locations report dissimilar dominating species of *Lactobacillaceae* representatives that are perhaps, in one way or another, attributed to the local population lifestyle bias.

The lack of scientific data regarding the probiotic microbiota of breast milk and infant feces characteristic of Bulgarian population brought about the base interest of the present study. Therefore, samples were collected from Bulgarian volunteers, breastfeeding mothers and their neonates. Basic information regarding any supplementation intake was requested, along with details about the pregnancy, birth mode, gender of the baby, etc. The samples were processed according to the requirements of lactic acid bacteria and the isolated strains were identified by the MALDI-TOF system. The MALDI-TOF MS method has a higher success rate than the polymerase chain reaction method in identifying the *Lactobacillus* species at the species level and, based on the reported literature results, it can be concluded that the method is applicable and accurate [28,29,30,31,32,33,34].

Inter-species and intra-species diversity was investigated by RAPD-PCR. Numerous research groups [35,36,37,38,39,40,41,42] have demonstrated the effectiveness of the method for this purpose. This approach requires unsophisticated execution, while the sample handing is simple and expeditious, and the results provide crucial information about the genetic variability in a group of species.

The studies published so far on the composition of the microbiota of human breast milk and the microbiota of the feces of infants in Bulgaria are extremely limited. On the other hand, lactobacilli of human origin have received much attention due to their potential health properties. They are commonly used as probiotics in foods and supplements and for pharmaceutical applications. Therefore, the aim of our study was to isolate, identify, and characterize some of the most common lactobacilli in samples of human breast milk and infant feces collected from a Bulgarian cohort, as well as to investigate the diversity among the strains of the most common newly isolated species. The obtained knowledge would provide information both for future fundamental studies on the distribution of lactobacilli in the Bulgarian population and for the selection of strains for further investigation of their probiotic potential. By maximizing the isolation of diverse lactobacilli, we can gain insight into diversity of the human milk microbiome depending on geographical location.

## 2. Materials and Methods

### 2.1. Samples

All the studies involving human participants were reviewed according to the Declaration of Helsinki and complied with all rules of bioethics. All the procedures concerning sample collection and analyses were approved by the Ethics commission of Sofia University “St. Kliment Ohridski” approval: №93-И-8≠1/24 January 2022 (№93-I-8≠1/24 January 2022). The participants/donors provided their written informed consent to participate in our study. We handled all the samples and personal data anonymously and published the results using unique codes. Mature breast milk and fecal samples were collected from fourteen healthy mothers from the Sofia region and their infants up to the first month after delivery. They donated one sample between 3 and 4 weeks from the beginning of the lactation period.

The following information was collected from the subjects: mode of delivery, infant’s type of feeding (exclusive breastfeeding, mixed feeding or only formula feeding), mother’s antibiotic and probiotic intake during pregnancy and lactation period and infant antibiotic and probiotic supplementation, etc. (Appendix A).

### 2.2. Sample Processing

Donors were supplied with sterile tubes for human milk and fecal collection tubes with spoon and screw cap and before sample collection, mothers were given written instructions for standardization of the sampling process. Proper milk samples were obtained through the following protocol: (1) the nipple was washed thoroughly with soap and water; (2) the first few drops of milk were discarded; (3) a total of 30 mL was collected in the respective collection tube, (4) samples were stored at 4 °C (not more than 24 h) until the transport to the laboratory. The protocol for fecal sample gathering was as follows: (1) fecal material (at least 2–3 g) was collected from the diaper with the special spoon of the respective collection tube; (2) samples were stored at 4 °C (not more than 24 h) until delivery to the laboratory.

In the current study, we used samples from 14 independent mother–newborn tandem pairs, 3 of which were provided by the Human Milk Bank, Sofia, Bulgaria, and 11 samples were from the volunteer mothers.

### 2.3. Isolation of Lactobacilli and Identification by MALDI-TOF-MS

Standard laboratory protocols were implemented for the isolation and identification of lactobacilli from human breast milk (HBM) and fecal samples. First, 9 mL of De Man, Rogosa, and Sharpe (MRS) broth medium (Oxoid Ltd., Hampshire, UK) supplemented with 0.05% L-cysteine was inoculated with 1 mL of fresh human breast milk or 200 mg of infant feces and cultivated for 48 h at 37 °C in anaerobic conditions. All the samples were enriched in MRS broth up to 24 h after the sampling time. Afterwards, the appropriate ten-fold dilutions were plated on MRS agar (Oxoid, Basingstoke, UK) supplemented with vancomycin (10 mg/L). The petri dishes were incubated under anaerobic conditions (GasPak™ EZ Anaerobe Sachets, Becton, Dickinson Company, Franklin Lakes, NJ, USA) at 37 °C for 48–72 h. Representative numbers of colonies randomly picked from the assayed medium were purified by streaking on the new dishes with MRS agar media. The colonies were selected according to their morphological characteristics (color, colony size, shape, etc.). The pure isolates were tested for their Gram reaction and catalase activity. After a microscopic examination of the cell morphology of the isolates, only rod-shaped ones were used for further analysis. Gram-positive, catalase-negative colonies with rod-shaped cells were selected as presumptive lactobacilli and were stored at −20 °C in MRS liquid medium, supplemented with 25% (*v*/*v*) of glycerol.

We used a matrix-assisted laser desorption/ionization–time of flight–mass analysis (MALDI-TOF-MS) (Bruker Daltonics, Billerica, MA, USA) for the direct identification of 68 preliminary selected pure isolates. A single overnight bacterial colony from each isolate were picked and transferred onto a polished steel MSP 96 target plate. The samples were covered with 1 µL of 70% formic acid and left to air-dry. Deposited samples were overlaid with 1 µL of a saturated cyano-4-hydroxycinnamic acid (HCCA) matrix solution (Bruker Daltonics). Unidentified strains were resubmitted using the extended protocol. Mass spectra were acquired using the microflex LT mass spectrometer (Bruker Daltonics) and analyzed with the research-use-only (RUO) software workflow and reference library MBT v. 4.1.100.

### 2.4. 16S rRNA Gene Sequencing

Selected strains were analyzed by 16S rRNA sequencing (Appendix A). Individual colonies from the strains identified by MALDI-TOF, as well as some marked as “no reliable identification” according to score values, were cultivated overnight in MRS broth. From each sample, DNA was extracted with Quick-DNA™ Fungal/Bacterial Miniprep Kit (Zymo Research Corp., Irvine, CA, USA) following the manufacturer’s instructions. DNA samples were subjected to Sanger sequencing. For the amplification of 16S rRNA genes, primers 27F (5′-AGAGTTTGATCMTGGCTCAG-3′) and 1492R (5′-TACGGYTACCTTGTTACGACTT-3′) were applied. PCR amplification was performed in a 25 μL volume and included the following: 50–70 ng of isolated genomic DNA, 1 µL of each primer with a concentration of 10 µM, and 12.5 µL of Supreme NZYTaq II 2x Green Master Mix (NZYtech, Lda, Lisboa, Portugal). The PCR program comprised the following: denaturation at 95 °C for 10 min, followed by 35 cycles of 94 °C for 30 s, 50 °C for 30 s, and 72 °C for 85 s, with the final extension of 7 min at 72 °C. The resulting PCR products (1500 bp) were purified using Agarose-out DNA Purfication Kit (EURx, Gdansk, Poland), according to the manufacturer’s instructions, and directly sequenced by Macrogen Inc. (Amsterdam, The Netherlands) on an automatic sequencer (Applied Biosystems Inc., Foster City, CA, USA) with the di-deoxy termination procedure in both directions using the universal primers 27F and 1492R. The obtained sequences were assembled and manually edited using the Vector NTI v. 10 software package. They were deposited in the GenBank of the National Center for Biotechnology Information (NCBI) database under accession numbers from PQ008844 to PQ008869 (https://www.ncbi.nlm.nih.gov/nuccore, accessed on 17 July 2024).

All retrieved sequences were used for the construction of the phylogenetic tree, applying the Neighbor-Joining method, Mega 6.0 program. The tree was constructed with 16S rRNA nucleotide sequences and the nearest high homology sequences that were obtained after a Blast search in the NCBI database.

### 2.5. Random Amplified Polymorphic DNA-PCR and Agarose Gel Electrophoresis (RAPD-PCR)

RAPD-PCR with single arbitrary primers was used for the molecular characterization of the dominant bacterial group identified as well as to assess the presence of genetic variation within the group. Strains belonging to the *L. rhamnosus* (*n* = 12), *L. paracasei* (*n* = 2) and *Lactobacillus zeae* (*n* = 1) species (Appendix A) were genetically characterized by RAPD-PCR analysis. For this purpose, PCR amplification reactions were performed in a 25 μL volume and included the following: 50–70 ng of isolated genomic DNA, 1 µL of each primer with a concentration of 10 µM, and 12.5 µL of Supreme NZYTaq II 2x Green Master Mix (NZYtech, Lda, Lisboa, Portugal). All RAPD-PCR primers and reaction conditions are listed in Appendix A. All PCR reactions were performed with a C1000 Touch Thermal Cycler (Bio-Rad Laboratories, Inc., Hercules, CA, USA). The obtained amplification products and DNA Ladder (peqGOLD 100 bp DNA Ladder Plus, VWR Int., Leuven, Belgium) were separated by electrophoresis on 2% agarose gel in 1xTBE buffer and stained with the GelGreen™ Nucleic Acid Gel Stain (Biotium, Fremont, CA, USA). The gels were documented with the ChemiDoc^TM^ Imaging System (Bio-Rad Laboratories, Inc., Hercules, CA, USA). The type strains *Lacticaseibacillus rhamnosus* ATCC 53103 and *Lacticaseibacillus paracasei* ATCC 334 were used as comparative controls to verify the primer specificity (Appendix A). At least two independent amplification reactions were performed for each primer.

## 3. Results

### 3.1. Isolation and Identification of Lactobacilli from Human Breast Milk and Infant Feces

A total of 68 isolates were obtained from fourteen pairs of mothers and their newborns, corresponding to the initial characteristics of lactobacilli as the object of primary interest in this study. The macro- and micromorphology of the strains were estimated by plate counting on the MRS agar. Most bacterial colonies appear white or translucent, generally round and smooth. Colonies with different morphologies were purified by streaking on new MRS agar plates. Microscope observations of the cell morphology of the isolates showed that the majority of them were represented by short rods or rods polymorphic in size, occurring singly or in short chains. All isolates that were rod-shaped, lacking catalase activity, and Gram positive were chosen for further characterization by MALDI-TOF-MS analyses. The obtained MALDI-TOF-MS profiles were compared to the reference spectra of the BioTyper database, and their similarity was expressed by score values. The color code illustrated the matching of the experimental MALDI-TOF-MS profile of the tested strain and the reference MALDI-TOF-MS profiles of the BioTyper database. The green, yellow, and red colors indicate the meaning of the score value and its interpretation as high confidence (2–3), low confidence (1.70–1.99), and no organism identification, respectively. A total of 29 lactobacilli colonies from breast milk and 39 from infant feces were identified by MALDI-TOF-MS. In total, five different genera of lactobacilli including *Lacticaseibacillus*, *Limosilactibacillus*, *Lactiplantibacillus*, *Levilactobacillus*, *Lactobacillus*, and the following eight different species belonging to the mentioned genus were identified: *Lacticaseibacillus rhamnosus*, *Lacticaseibacillus paracasei*, *Limosilactibacillus reuteri*, *Limosilactibacillus fermentum*, *Lactiplantibacillus plantarum*, *Levilactobacillus brevis*, *Lactobacillus gasseri*, *Lactobacillus zeae*. The isolates were identified with a score value of between 1.73 and 2.51 (Appendix A); up to 42 of them (61.8%) had a score ≥ 2.0, 14 (20.6%) had a score ≥ 1.9, and 12 isolates (17.6%) had a score ≥ 1.7 (Appendix A).

The results obtained by MALDI-TOF showed that the most frequently detected species was *L. rhamnosus*, which was found in five HBM samples and five fecal samples and represented up to 34% of all isolates. The second most abundant species was *L. fermentum*, constituting 26% of the isolated strains and found in two breast milk samples and five fecal ones. *L. paracasei* and *L. reuteri* followed, each corresponding to 15% of the detected species. *L. paracasei* was isolated from two breast milk samples and four fecal samples, while *L. reuteri* from one HBM and three from feces. *L. plantarum* was detected in two HBM probes and one fecal sample, while *L. gasseri* was detected in only one breast milk sample. Although they were single strains, the presence of the *L. brevis* and *L. zeae* speies was found. They were both isolated only from the HBM samples (Figure 1, Appendix A).

In our study, we demonstrated that more than 50% of all the detected species were found in both samples of the tandem mother–infant pairs (Table 1). For instance, *L. fermentum* was discovered in probes El (BM) and El (F), as well as Y (BM) and Y (F). In the same manner, M (BM) and M (F), N (BM) and N (F), KV (BM) and KV (F), all contained isolates of *L. rhamnosus.* The strains of *L. paracasei* followed the same principle by being presented in M (BM) and M (F), KV (BM) and KV (F). *L. plantarum* was isolated from the tandem pair Sv.

### 3.2. 16S Ribosomal RNA Gene Sequencing

In order to confirm the obtained results by MALDI-TOF and to get more profound information regarding the abundance of lactobacilli in the Bulgarian human breast milk and neonate fecal samples, 26 isolates were selected for the proceeding 16S rRNA sequencing (Appendix A). The strains were chosen based on sample type (almost identical number of strains originating from milk and fecal samples), presence of the species in both subjects in a mother–infant tandem pair, for example, M1-1, M1F-2, KV1-1, KV1F-4, KV1F-5, N1-1, N1F-2, El1-2, El1F-3, Sv1-1, and Sv2F-1. Some strains found only in one member of the tandem pair were also included in the sequencing analysis (Zdr, Iv, Hr, L1, 40/2, Y1F). These isolates were used to reveal potentially individual differences among the donors, like Zdr1F-1 and Y1F-3, both belonging to *L. fermentum*. In the same principle, we selected the *L. rhamnosus* isolates, L1-1, Hr1F-1, and 40/2F-1. Also, we sequenced some isolates with the same origin in order to determine the presence of potential species diversity (1/24-1 and 1/24-2, Iv1-1 and Iv1-2) in a single sample. Lastly, the only representatives of *L. gasseri* (1/24-1, 1/24-2), *L. brevis*, and *L. zeae*, (5/24F-1 and 40/2F-2, respectively) were incorporated because of their non-repetitiveness and exclusivity.

The PCR amplification resulted in PCR products of approximately 1500 bp. For the identification and phylogenetic determination, the obtained sequences and their retrieved nearest relatives from the BLASTn search into the GenBank standard database non-re-dundant-nr/nt were used. The obtained results identified 26 different sequences, for which the identity varied from 99.56% (21 sequences) to 100% (3 sequences) [43]. Only two had a lower identity of about 99.30% (Appendix A).

To strengthen the MALDI-TOF data, we performed the identification of 26 selected strains by 16S rRNA sequencing. The combined results from MALDI-TOF mass spectrometry and phylogenetic analysis demonstrated that the 26 obtained sequences belonged to the five genera, *Lacticaseibacillus*, *Lactobacillus*, *Limosilactibacillus*, *Lactiplanplantibacillus*, and *Levilactobacillus*. At the same time, the phylogenetic tree (Figure 2) showed the presence of the following four clusters: the first cluster (Cluster I) consisted of 12 different *L. rhamnosus* strains and two *L. paracasei* strains. The second cluster (Cluster II) was divided into two subclusters, which contained one *L. brevis* strain, two *L. plantarum* strains, and one strain partially sequenced but closest to the species *L. pentosus* strain (strain El2-1). Notably, MALDI-TOF mass spectrometry identified El2-1 as *L. plantarum* with a score value of 2.02. A third cluster (Cluster III) was also formed with two subclusters, including two *L. reuteri* strains and four *L. fermentum* strains. Cluster IV included two strains of *L. gasseri*. MALDI-TOF-MS identification also showed the presence of the *L. zeae* strain (strain 40/2F-2), but 16S rRNA sequencing did not confirm it. The score value of MALDI-TOF-MS identification of the strain was 2.06, and the second-best match was *L. rhamnosus* with score of 1.90. We assumed that MALDI-TOF-MS may generate spectra that were too similar for *L. zeae* and *L. rhamnosus* as 16S rRNA studies have shown that these two species are closely related within the *L. casei* group [44]. The applied approach by 16S rRNA sequencing and MALDI-TOF mass spectrometry in this study supports the idea of rich intragenic diversity in the samples with domination of the *Lacticaseibacillus* genus.

### 3.3. Strain Diversity of L. rhamnosus (Cluster I Strain Group)

Fifteen strains, belonging to cluster I, and the two type strains of *L. paracasei* ATCC 334 and *L. rhamnosus* ATCC 53103 were subjected to RAPD-PCR analysis in order to define any strain diversity or similarity within the bacterial species that had been established as most frequently isolated. The genetic fingerprinting was performed by using four short-length single primers, RAPD-04, 80A_RAPD, 80B_RAPD_M13, and 80C_RAPD_OPT-14, as described in Appendix A. Primer RAPD-04 was selected for its superior discriminative power to five other RAPD primers (opp-07, opp-08, opp-09, opp-14, and rapd-06). On the other hand, primers 80A_RAPD, 80B_RAPD_M13, and 80C_RAPD_OPT-14 were appointed based on the literature data [37,45].

The obtained fingerprinting profiles of the strains from cluster I manifested genetic similarity, not only among the isolates originating from one mother–child tandem pair but also from the different donors and sample types (Figure 3). For example, a mother– child pair designated in this study as KV showed both a common *L. rhamnosus* strain in the breast milk (isolate KV1-1) and fecal samples (isolate KV1F-4), and one additional strain found only in the child (KV1F-5), while the mother carried an additional *L. paracasei* strain (KV2-6). Similarly, a common strain of *L. rhamnosus* was found in the tandem pair designated as M (isolates M1-1 and M1F2), and, again, the mother hosted an additional *L. paracasei* strain (isolate M1-5). Notably, isolates M1-1 and M1F2 showed RAPD profiles that were nearly identical to the type strain *L. rhamnosus* ATCC 53103. Also, the two *L. paracasei* isolates included in this analysis clustered together with the type strain *L. paracasei* ATCC 334, well separated from all the *L. rhamnosus* cultures. Closely related or identical *L. rhamnosus* strains were found in the fecal samples from different children (isolates 40/2F1, 40/2F2 and Hr1F1) or in the breast milk samples from different mothers (isolates N1-1 and KV1-1). On the other hand, some *L. rhamnosus* isolates had a unique profile for a single donor such as isolates L1-1, Iv1-1, and Iv1-2 from the breast milk samples and KV1F-5 and N1F2 from the fecal samples. Within the 15 analyzed isolates from cluster I, 9 different RAPD profiles were observed (7 for *L. rhamnosus* and 2 for *L. paracasei* isolates), suggesting a high abundance of different genotypes and strain diversity in the samples (Figure 4).

## 4. Discussion

The presence of valuable ingredients in breast milk, including beneficial microbiota, depends on multiple factors, mainly maternal lifestyle, conditions, geography [22], etc. The microbial characteristics of infant gut microbiota that, even though originating during pregnancy, are defined by another set of factors, such as birth mode, gestational age, hospital environment, and the occurrence of formula feeding [46]. However, the most significant impact on the neonate trophic and metabolic characteristics [6], immune system development, and infection susceptibility is owed to breastfeeding manners. Staphylococci, streptococci, corynebacteria, propionibacteria, and some lactic acid bacteria are among the traditionally reported bacteria [6,45,47] in human breast milk. The probiotic microflora of breast milk has been of great scientific interest during the recent years, its origin, distribution and transmission from the mother to the child, along with the different species, strains and their metabolic profile, which determines their potentially beneficial functions [6].

Geographical location plays a pivotal role in the microbiota proportions and has been the main factor among many studies [18,22,23,25,26,27]. Ding et al. demonstrated the prevailing tendency of *L. reuteri* and *L. gasseri* in breast milk among the Chinese population by sampling 89 healthy women from 11 different regions [23]. *L. gasseri* was also documented as the most frequently found species in human milk and neonate feces from the previous *Lactobacillus* genus in Spain [18,26] and Ireland [48]. Albesharat et al. [47] reported *L. plantarum*, *L. fermentum*, and *L. brevis* as the main *Lactobacillaceae* representatives not only in Syrian breast milk but also infant and mother’s feces, as well as local fermented foods. In Germany and Austria, the dominating lactobacilli species were found to be *L. salivarius*, followed by *L. fermentum*, according to Soto et al. [24]. Over 30% of the isolates in our study belonged to the *L. rhamnosus* species, a finding suggesting that this species is potentially the main representative of cultivable Bulgarian breast milk lactic acid microbiota. The lack of previously reported data on this matter additionally reinforces the significance of the obtained results but also underlines the need for further work in the field. The only data published so far concerning the investigation of lactobacilli with origin from human breast milk in Bulgaria were from Mollova et al. [49]. They studied the genomic and phenotypic aspects of the *L. plantarum* PU3 strain that showed potential as a probiotic agent [49]. *L. fermentum* was the second most encountered isolate in human milk and neonate feces, a result also manifested in the works of Albesharat et al. [47] and Soto et al. [24]. Moreover, the prevalence of *L. rhamnosus* among other lactobacilli in the infant fecal samples, followed by the *L. fermentum* representatives was also observed by Ahrné et al. [50] in Swedish children before weaning. Mitsou et al. reported predominance of *L. rhamnosus* in stool samples of healthy Greek neonates, and *L. paracasei* as the third most abundant lactobacilli representative [25]. Nikolopoulou et al. investigated breast milk samples collected from 100 healthy women in Greece, where 26 samples were colostrum and 74 samples were mature breast milk. The presence of the genus *Lactobacillus* was identified in 46.2% of colostrum and 24.3% of mature breast milk [51]. Eventually, different geographic regions display unique microbiological profiles, probably related to lifestyle, climate, customs, or genetics. However, a notable factor for the reported results is the individuality of the participants in similar studies, and the normal deviations that occur in the microbiota of a particular donor. Therefore, the number of subjects is a determining factor for reporting the most accurate information.

Recently, a correlation between the infants’ and adults’ gut microbiome was reported, indicating that the infants’ gut is initially colonized by bacteria originating from either breast milk or the environment [52]. In our study, similar dynamics of *Lactobacillaceae* microbiota are displayed in the Bulgarian human breast milk and neonate fecal samples, whilst more than 50% of all the detected species were found in both samples of the mother–infant tandem pairs (Table 1). This microbial succession from mother’s milk to the infant’s gut exemplifies the notability of breastfeeding and its crucial role in gut microbiota development in infants. However, Zhang et al. reported a lack of correlation between lactobacilli isolated from breast milk and infant feces in their research [27]. We also detected some strains isolated only from HBM or feces. In our study, we revealed, for the first time, the most frequently found lactobacilli isolated from the breast milk of healthy Bulgarian mothers as well from the fecal samples of their infants. We found out that the predominant rod-shaped lactobacilli in both types of tested samples (HBM and feces) were *L. rhamnosus*, *L. fermentum*, *L. paracasei*, and *L. reuteri*. Moreover, we isolated *L. plantarum* and *L. gasseri* from HBM and *L. plantarum*, *L. zeae* and *L. brevis* from feces according to MALDI-TOF-MS identification, even though they represented small percentages of the total amount of the identified species. Specifically, we succeeded in identifying six species from HBM and seven species from feces. Interestingly, despite the diversity of species, combination of not more than two species from the different samples was isolated. Various factors such as the mother’s lifestyle, lactation period, and residential location may contribute to the presence of those species in the tested samples.

Based on traditional culturing, the main taxa reported in human breast milk in the literature were *L. casei*, *L. salivarius*, *L. plantarum*, *L. fermentum*, *L. rhamnosus*, *L. gasseri*, *L. paracasei*, *L. oris* [18,23,24,27,47]. In the infant feces, the *Lactobacillus* species most frequently isolated and detected were *L. brevis*, *L. fermentum*, *L. reuteri*, *L. rhamnosus L. plantarum*, *L. mucosae*, *L. ruminis* [26,47,53,54].

Given the existing two main hypotheses [35] regarding the origin of breast milk microbiota and based on the results we obtained, we could say that both hypotheses provide a tendency of accuracy considering the cases, in which the presence of one strain in both a breast milk and a fecal sample was surveilled. Therefore, no conclusion on the breastmilk microbiota origin could be provided. Additional experimental work, including mother fecal samples for reference, along with more participants for increased factuality would shed light on the accuracy of these hypotheses.

In their study, Martín et al. [18] reported a lack of strain similarity between the human milk and breast skin or neonate fecal *L. gasseri* isolates and suggested that the vast majority of lactic acid bacteria in the breast milk have an endogenous origin. No resemblance between the infant fecal lactobacilli and vaginal isolates from the respective mothers was observed, which raised the question whether mother vaginal microbiome impacts the naturally born neonate gut microbiota [18]. In contrast, Martin et al. [55] demonstrated cases of strain homogeneity of *L. gasseri*, *L. fermentum*, *L. salivarius* and other lactobacilli in the breastmilk and infant fecal samples. The data displayed in our study corresponded to similar findings. For that reason, our study additionally confirms the existing transfer of potentially probiotic microbiota from the breast milk to the neonate gut. In the present work, MALDI-TOF-MS correctly identified the most frequently isolated species of lactobacilli from the human breast milk and infant feces samples, although the number of some isolated species was low. The applied approach by culturomics, which combined the classical cultivation method with MALDI-TOF mass spectrometry and 16S rRNA sequencing to identify bacterial isolates from complex ecosystems, supports the idea that pure bacterial isolates serve as a main tool in microbiome research. The culturomics methodology is widely applied, particularly for investigating the microorganisms in the gut, vagina, mouth, and urinary tract [56,57,58,59,60]. On the other hand, only a few studies have reported on their application of human milk microbiota [30,61,62,63]. The correlation between 16S rRNA and MALDI-TOF-MS-based identification of the strains selected by us indicates that the culturomics strategy is reliable for the identification of lactobacilli.

The obtained results revealed a significant genus diversity in the samples with domination of the *Lacticaseibacillus* genus. Although the number of analyzed isolates is not sufficient to draw conclusions about the strain distribution in the mother–child tandem pairs or in a larger cohort of volunteers, the data obtained show a high abundance of *L. rhamnosus* strains in this ecosystem.

*L. rhamnosus*, *L. paracasei*, and *L. casei* are part of the *L. casei* group that possesses certain metabolic, morphological, and genetic similarity, making their differentiation complicated [37]. To investigate the strain diversity among the most common species, *L. rhamnosus*, in the samples we examined, we applied RAPD-PCR to the isolates positioned in Cluster I in the Neighbor-Joining phylogenic tree that were also all belonging to the *L. casei* group (thirteen *L. rhamnosus* strains and two *L. paracasei* strains). RAPD is a useful tool for generating genomic information of newly isolated strains. This technique uses a single short (around 10 bases in length) primer [42] that randomly hybridizes to a location of DNA, which gets amplified and after electrophoretic separation produces a fingerprint profile to be utilized in demonstrating genetic variability between two or more isolates [38]. A number of studies demonstrate the effectiveness of this analysis, both for distinguishing some species from the *L. casei* group and for studying intraspecies genetic diversity.

In their study, Mahenthiralingam et al. thoroughly investigated the discriminative ability and the reproducibility of the RAPD-PCR by performing RAPD fingerprinting using a set of 100 primers [38]. Even though this method was noted for its low reproducibility [46], the obtained results demonstrate high reproducibility, the proper clustering of *L. casei* and *L. acidophilus* group representatives, and a clear differentiation between the genetically distant strains of lactobacilli and different members of the same cluster [41]. Many genetic variability determining methods were compared in the study of Jarotski et al. with PCR-RAPD being described by the authors as one of high differential strength. They tested the distinguishing properties of four RAPD primers and discovered that 80A_RAPD, 80B_RAPD_M13, and 80C_RAPD_OPT-14 effectively differentiated between 30 lactobacilli strains belonging to the *L. casei* group [37]. These and other studies [45,46] proving the capacity of RAPD-PCR analysis for genetic variability detection determined its implementation in the following study of genetic diversity of species belonging to the *L. casei* group.

Using RAPD-PCR analysis in our study, we revealed a high strain diversity within the studied isolates of *L. rhamnosus* (seven different RAPD profiles) and showed the applicability of this method to distinguish between *L. rhamnosus* and *L. paracaseae* isolates thus confirming the effectiveness of this method both for studying intraspecies diversity and for distinguishing some species from the *L. casei* group.

## 5. Conclusions

In the recent study, the polyphase approach including reliable techniques such as MALDI-TOF mass spectrometry and 16S rRNA were applied to identify cultivable lactobacilli population isolated from the HBM and infant feces samples (Appendix A). On the other hand, to provide evidence for strain diversity between the most common species of *L. rhamnosus* tested, RAPD-PCR analysis was conducted. The isolated cultures were affiliated to five different genera of lactobacilli including *Lacticaseibacillus*, *Limosilactibacillus*, *Lactiplantibacillus*, *Levilactobacillus*, and *Lactobacillus* and seven different species including *L. rhamnosus*, *L. paracasei*, *L. reuteri*, *L. fermentum*, *L. plantarum*, *L. brevis*, and *L. gasseri.* Our data revealed that *L. rhamnosus* was the predominant species isolated from the tested samples of HBM and infant feces from the Bulgarian donors. Through RAPD analysis of the strains belonging to the most frequently isolated species, *L. rhamnosus*, the presence of significant strain diversity among this species was shown.

Our pilot study revealed the cultivable, most frequently isolated, rod-shaped lactic acid bacteria probably indigenous to the breast milk of Bulgarian mothers as well as for the gut microbiota of their infant. The results obtained from this study complement the available information on the distribution of cultivable lactobacilli in the breast milk and in the infant feces, which is scarce for Europe as a whole and in particular for Bulgaria, and could be used in the comparative studies aimed at clarifying a number of scientific questions related to the origin of the microbiota of breast milk and its transfer to the GIT of newborns, as well as the geographical differences in its composition. On the other hand, the isolated cultivable strains of lactobacilli could be a good source for the development of probiotic preparations adapted to the Bulgarian population. We believe that this research contributes to the development of the culturomics strategy (culture methods in combination with MALDI-TOF-MS and 16S rRNA sequencing) for the isolation and identification of lactobacilli from breast milk and infant feces and can enhance the ability to analyze the human milk microbiota in vitro, providing a more detailed understanding of the microbial communities in this complex environment.

## Figures and Tables

**Figure 1 microorganisms-12-01839-f001:**
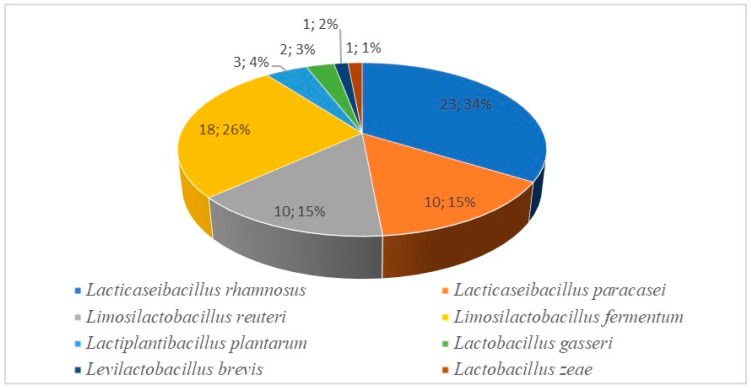
Species distribution (number and percentage) according to the MALDI-TOF MS identification.

**Figure 2 microorganisms-12-01839-f002:**
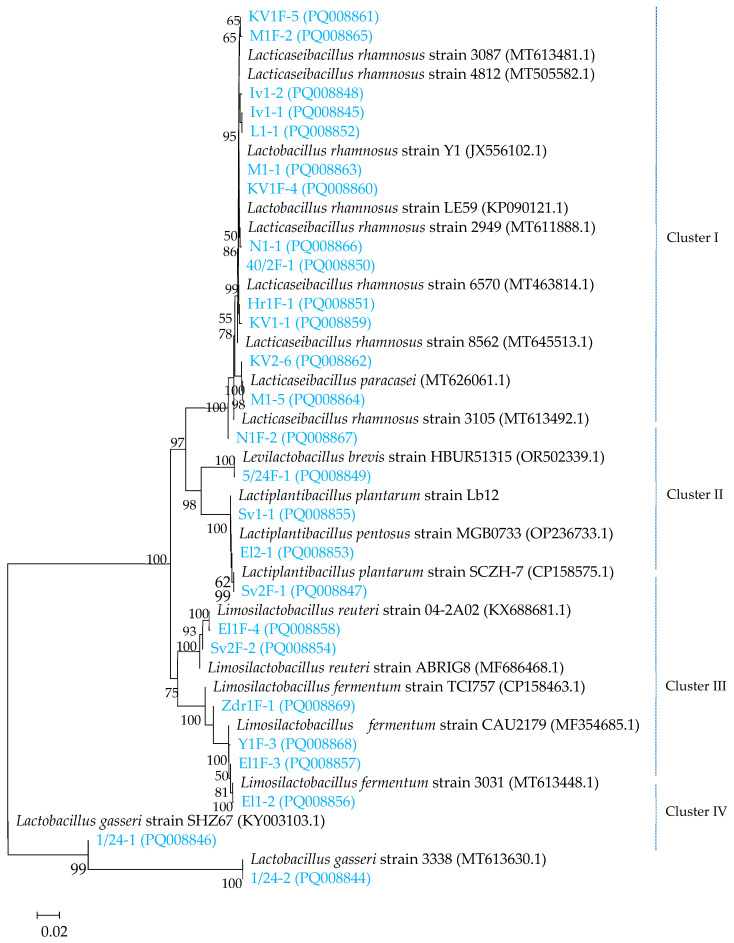
Neighbor-joining phylogenetic tree, of 26 lactobacilli strains (blue-colored) based on 16S rRNA gene sequences.

**Figure 3 microorganisms-12-01839-f003:**
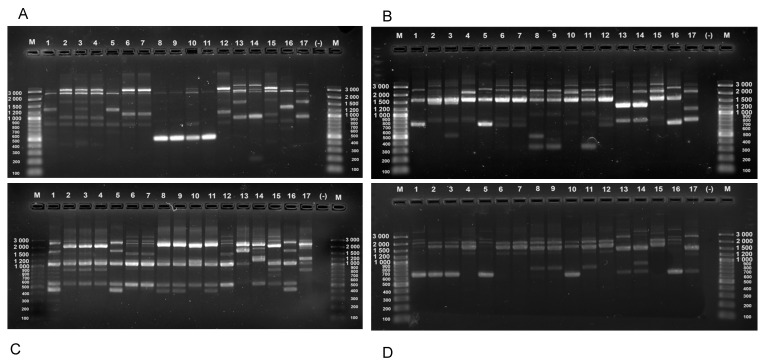
RAPD patterns of *L. rhamnosus* and *L. paracasei* strains obtained with primers (**A**) RAPD-04, (**B**) 80A_RAPD, (**C**) 80C_RAPD_OPT-14 and (**D**) 80B_RAPD_M13. M-DNA Ladder, Lane 1 to 17 corresponding to the strain numbers: M1F-2, Iv1-2, Iv1-1, L1-1, M1-1, KV1F-4, N1-1, 40/2F-1, 40/2F-2, KV1F-5, Hr1F-5, KV1-1, KV2-6, M1-5, N1F-2, *L. rhamnosus* ATCC 53103, *L. paracasei* ATCC 334, (-)—negative control.

**Figure 4 microorganisms-12-01839-f004:**
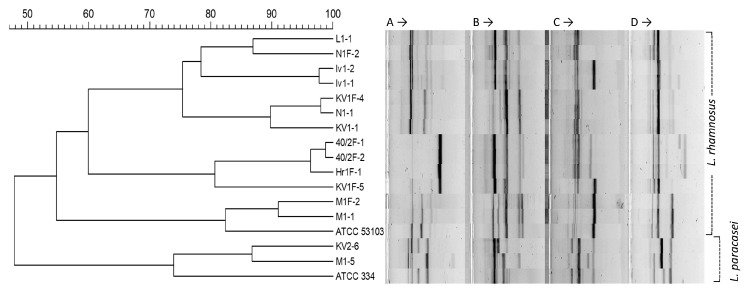
Clustering of *L. rhamnosus* and *L. paracasei* isolates according to their RAPD profiling with primers (**A**) RAPD-04, (**B**) 80A_RAPD, (**C**) 80C_RAPD_OPT-14 and (**D**) 80B_RAPD_M13. The tree was generated using the unweighted pair group method with arithmetic mean (UPGMA) based on calculating the Pearson’s Correlation coefficient.

**Table 1 microorganisms-12-01839-t001:** Distribution of the same lactobacilli species in a mother–infant pair.

Species	Strains	Sample Information	
		ID of Tandem Pair Sample Mother-Infant	Origin of Sample
*L. rhamnosus*	M1-1; M2-3	M	BM
	M1F-1; M1F-2; M1F-3; M1F-7; M1F-8; M2F-4; M2F-5		F
	KV1-1; KV1-2; KV1-3	KV	BM
	KV1F-4; KV1F-5; KV2F-3		F
	N1-1	N	BM
	N1F-1; N1F-2		F
*L. fermentum*	EI1-1; EI1-2; EI-3; EI1-4; EI1-5; EI1-6	EI	BM
	EI1F-2; EI1F-3		F
	Y1-1; Y1-2	Y	BM
	Y1F-3		F
*L. paracasei*	M1-4; M1-5; M2-2	M	BM
	M1F-6		F
	KV2-6	KV	BM
	KV2F-2; KV2F-7		F
*L. plantarum*	Sv1-1	Sv	BM
	Sv2F-1		F

BM—Breast milk; F—Infant Feces.

## Data Availability

Data are contained within the article and Appendix A.

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
