# Peer review of "Identification and Characterization of Human Breast Milk and Infant Fecal Cultivable Lactobacilli Isolated in Bulgaria: A Pilot Study"

_microorganisms, 2024, doi:10.3390/microorganisms12091839_

Round 1

Reviewer 1 Report

Comments and Suggestions for Authors

The paper is well-written, with a suitable background that reflects the current state of research in this field. However, the methods used may not capture all cultivable lactic acid bacteria present in human milk. Moreover, the author states, "we revealed for the first time the core lactobacilli microbiota of breast milk of healthy Bulgarian mothers as well as the core lactobacilli microbiota isolated from infant feces. Firstly, this is not the first study regarding lactobacilli in human milk from Bulgarian mothers (which are also cited in this manuscript). Secondly, to reveal the “core” lactobacilli microbiota, I believe a non-culturable approach, like NGS (Next-Generation Sequencing), would be more appropriate to obtain a comprehensive view of the milk microbiota. Quite a few clinical approaches could affect the outcomes derived from the “culture technique”. Additionally, the results and discussion mention and compare the differences in lactic acid bacteria distribution across different countries. However, generally speaking, to compare the distribution proportions of dominant or probiotic strains, the NGS method should be used to obtain a complete picture, although many factors influence the distribution of probiotics in human milk. Nevertheless, this pilot study has provided interesting data regarding the Lactic Acid Bacteria profile isolated from human milk and infant fecal samples.

specific points:

 1.        Although some lactic acid bacteria (LAB) may have natural resistance to vancomycin, many LAB are sensitive to it. Therefore, please explain the rationale for adding vancomycin to the MRS medium. What is the basis for the vancomycin concentration (10 mg/L) used? Does this concentration also inhibit the growth of most LAB?

2.        How did you determine that the LAB you isolated are the dominant bacterial group? Especially given that you used vancomycin; how can you be sure that you have isolated all culturable probiotics? Please clarify and provide the reasons or evidence.

3.        If you want to compare the dominant bacterial species, the NGS method should be used to obtain a comprehensive view. Please clarify.

4.        Although Bifidobacterium is not classified as LAB, it is generally present in significant numbers in human milk and infant feces and is important for host health. Why did you not intend to isolate or identify Bifidobacterium? Was it because you did not specifically isolate Bifidobacterium, or did you not identify it? Please clarify.

5.        From your bacterial isolation method, it is known that you directly inoculated 1 ml of fresh human milk into MRS medium. This method may not be able to culture all LAB present in human milk. Did you mix the sample beforehand? Did you centrifuge the milk sample? In general, to obtain and harvest all culturable bacterial strains from a milk sample, some research will centrifuge the sample, collect the pellet, and retain a small portion of the milk for bacterial isolation. Please clarify.

6.        In the results and discussion section, you discussed the distribution proportions of probiotics in human milk from various regions. However, it is unclear whether these proportions were obtained through isolation methods or NGS methods. It is recommended that the authors include this information; otherwise, the comparison is meaningless. Isolation methods can only provide a partial view, and only NGS or other specialized methods can obtain the complete human milk microbiota. Please revise or clarify.

7.        Line 349-351: The authors mentioned that "we revealed for the first time the core lactobacilli microbiota of breast milk of healthy Bulgarian mothers as well as the core lactobacilli microbiota isolated from infant feces." Firstly, this is not the first study regarding lactobacilli in human milk from Bulgarian mothers; secondly, to reveal the core lactobacilli microbiota, using a culturable approach is not appropriate. Please clarify.

Author Response

Generally

Dear Reviewer,

Thank you very much for your valuable review comments. According to the design and methods used in our study we chose them after a deep literature research and on the other hand we tried different approaches before the main experiments. We have been used different modifications of culture media (with and without L-cysteine, with GOS, low/high concentration of vancomycin) also seeding directly from the breast milk samples, centrifugation, etc. Yes, you are right that culturable methods may not capture all cultivable LAB presented in the breast milk or infant feces, but this is valid for all other types of samples. This is the limitation of such methods. Nevertheless, there is no other method to isolate pure cultures, identified to the species level and characterized them and to reveal the strain diversity. The novel methods like NGS and WGS are the best way to obtain more information about the whole microbiota of the samples or to characterize the strains, but more or less sometimes you cannot receive information about the identification of the LAB at the species level with NGS. We done in parallel NGS of all samples (procaryotic and eukaryotic microbiome) but we are still not ready with the interpretation of the data. Our work is part of ongoing project and we are obliged to publish the results on different steps to go ahead with the finances. So, I hope that in the coming months we can publish NGS data in comparison with main component consist of breast milk (lactose, protein, microelements and so on). The aim of the current study was to find out the most common lactobacilli in Bulgarian samples of human breast milk and infant feces in the tested Bulgarian cohort from a relatively narrow region (Sofia region). Our study is the first one, because there are no other published data from Bulgaria in that research area. Our colleagues that we cited, have been worked only with one strain L. plantarum isolated from human breast milk. There were no data concerning more samples of breast milk in combination with infant feces from different tandem pairs. That’s way we are the first who reveal the most common rod-shaped LAB, isolated from such matrices in Bulgaria. Our further aim is not only to isolate but to investigate the newly isolated strains for their probiotic potential. On the other hand, our data confirm the dominant rod-shaped LAB isolated from mentioned type of samples. Therefore, the achievements of our study are that we revealed for the first time five different genera of lactobacilli including Lacticaseibacillus, Limosilactibacillus, Lactiplantibacillus, Levilactobacillus, Lactobacillus and seven different species: L. rhamnosus, L. paracasei, L. reuteri, L. fermentum, L. plantarum, L. brevis, and L. gasseri. Based on traditional culturing, the main taxa reported in human breast milk in the literature were L. casei, L. salivarius, L. plantarum, L. fermentum, L. rhamnosus, L. gasseri and L. paracasei (Albesharat et al., 2011; Martín et al., 2012; Soto et al., 2014; Ding et al., 2019). In infant feces, the Lactobacillus species most frequently isolated and detected were L. brevis, L. fermentum, L. reuteri, L. rhamnosus, and L. plantarum ((Solís et al., 2010; Martín et al., 2012; Jost et al., 2014; Murphy et al.,2017)). We confirmed that information. Overall, the diversity and number of rod-shaped lactobacilli in breast milk and infant feces varies with different geographical locations and nationalities (lifestyle, food diet etc.).

Comment 1 and Comment 2: Although some lactic acid bacteria (LAB) may have natural resistance to vancomycin, many LAB are sensitive to it. Therefore, please explain the rationale for adding vancomycin to the MRS medium. What is the basis for the vancomycin concentration (10 mg/L) used? Does this concentration also inhibit the growth of most LAB? How did you determine that the LAB you isolated are the dominant bacterial group? Especially given that you used vancomycin; how can you be sure that you have isolated all culturable probiotics? Please clarify and provide the reasons or evidence.

Response 1/2: It’s a really interesting remark. We decided to use MRS supplemented with 10 mg/L vancomycin after a long period of optimization of the most appropriate medium for isolation of rod-shaped lactobacilli from our samples from human breast milk and infant feces. In the very bigging we started to use only MRS agar medium, but unfortunately, we detected mainly cocci and it was difficult to select lactobacilli. So, we optimized the process – first we conducted 48 hours of enrichment in MRS supplemented with L-cysteine, after that we plated on MRS agar supplemented with vancomycin. On the basis of literature data, we tried several concentrations of vancomycin and we established that 10 mg/L was the higher concentration that we can use as a supplement (we tested the concentration with different type strain, including L. acidophillus as a presumptive sensitive species). Using MRS with vancomycin meanwhile we isolated also vancomycin-resistant enterococci. Valérie Coeuret et al., 2003,  cited Hartemink et al. 1997, who developed a new selective medium, LAMVAB (Lactobacillus anaerobic MRS with vancomycin and bromocresol green), for the isolation of Lactobacillus species also included vancomycin. Firstly, they used it to isolate lactobacilli from feces (in which they are present in small numbers), and then successfully for various species of lactobacilli from dairy products. The medium is highly selective, due to its low pH and the presence of vancomycin (10 mg·L–1) However, this medium remains the most specific medium to date described for lactobacilli. They also noted that some Lactobacillus species, such as Lactobacillus delbrueckii spp. bulgaricus, and some strains of L. acidophilus are vancomycin-sensitive. Also, Sakai et al, 2010 used M-RTLV agar media (10 mg/L vancomycin) as a a novel selective medium to distinguish Lactobacillus casei and Lactobacillus paracasei from Lactobacillus rhamnosus. They established that the viable counts of L. fermentum, L. plantarum and L. reuteri on M-RTLV agar were equivalent to those on MRS agar. They detected growth on M-RTLV agar of different species of ex-Lactobacillus genus like L.sakei, L.fermentum, L. reuteri, L. casei, L. paracasei, L. rhamnosus and L. plantarum. As I mentioned based on traditional culturing, the main taxa reported in human breast milk and infant feces in the literature were L. casei, L. salivarius, L. plantarum, L. fermentum, L. rhamnosus, L. gasseri, L. paracasei, L, reuteri, L. brevis, L. ruminis, L. mucosae and L. salivarius. So, we also succeed to detected most of that species and that’s way we believed that our method is appropriate for isolation of the most frequently presented rod-shaped lactobacilli in the tested samples. Of course, we cannot be sure that we have isolated all the presumptive probiotic lactobacilli. In the manuscript, we make no such claim. In addition, in further studies we will check the probiotic potential of the lactobacilli isolated by us. In our case, perhaps the term dominant microflora is may be not the most accurate. Rather, we mean that these are the most frequently isolated species from the analysis of the samples we examined, so we changed it in the main text.

Comment 3: If you want to compare the dominant bacterial species, the NGS method should be used to obtain a comprehensive view. Please clarify.

Response 3: As I mentioned in my general comment, the novel methods like NGS and WGS are the best way to obtain more information about the whole microbiota of the samples or to characterize the strains. We done in parallel NGS of all samples and in our future article we will compare the dominant/core bacterial and eucaryotic species in the tested sample. At this stage our aim was to detect the most frequently isolated lactobacilli from Bulgarian samples and to compare them with the data from other geographical regions, as well as to use the newly isolated strain for further analyses like strain diversity and probiotic potential. We revised in the main text “dominant/core bacterial species” with “the most frequently isolated and detected rod-shaped lactobacilli”. Many researchers noticed that there was a connection between geographic region and the main microbiota. There are assumptions that strains isolated from the relevant geographic location are best adapted to the microbiota of the local population and have the potential to be used as probiotics specifically for these geographic areas.

Comment 4: Although Bifidobacterium is not classified as LAB, it is generally present in significant numbers in human milk and infant feces and is important for host health. Why did you not intend to isolate or identify Bifidobacterium? Was it because you did not specifically isolate Bifidobacterium, or did you not identify it? Please clarify.

Response 4: Thank you for your question. In our future analyses we intend to isolate and identified species from genus Bifidobacterium also. As a strict anaerobic species isolation of Bifidobacterium species is not so easy, we are in process to optimized it. Our observation for the moment revealed that generally we detected Bifidobacterium species mainly in the fecal samples and not so often in breast milk ones. Will be really nice to have more information not only for Bifidobacterium species but also for other strict anaerobic bacteria presented.

Comment 5: From your bacterial isolation method, it is known that you directly inoculated 1 ml of fresh human milk into MRS medium. This method may not be able to culture all LAB present in human milk. Did you mix the sample beforehand? Did you centrifuge the milk sample? In general, to obtain and harvest all culturable bacterial strains from a milk sample, some research will centrifuge the sample, collect the pellet, and retain a small portion of the milk for bacterial isolation. Please clarify.

Response 5: We collected about 30 ml of breast milk samples. Before the enrichment in MRS broth we mix each sample very well. In the beginning of our experiments, we tried to use different methods like direct plating on MRS, centrifugation of 5-10 ml of breast milk, but the pellet consisted a lot of cocci. In parallel we seeded breast milk samples on Columbia agar with 5% sheep blood and revealed that there were about 104-105 cfu/ml, and after centrifugation there were really a lot and was verry difficult to distinguished and selected the lactobacilli. This is also one of the reasons that we considered to use a vancomycin and step of enrichment of 1 ml. You are absolutely right that after concentration there is a better chance to reach potentially more species, but practically without of selective factor often is impossible. It’s a good idea to try in the future new optimization.

Comment 6: In the results and discussion section, you discussed the distribution proportions of probiotics in human milk from various regions. However, it is unclear whether these proportions were obtained through isolation methods or NGS methods. It is recommended that the authors include this information; otherwise, the comparison is meaningless. Isolation methods can only provide a partial view, and only NGS or other specialized methods can obtain the complete human milk microbiota. Please revise or clarify.

Response 6: We revised and clarified in the main text that we compared in our discussion the distribution of the detected strain according to the data received with the culturable-dependent methods.

Comment 7: Line 349-351: The authors mentioned that "we revealed for the first time the core lactobacilli microbiota of breast milk of healthy Bulgarian mothers as well as the core lactobacilli microbiota isolated from infant feces." Firstly, this is not the first study regarding lactobacilli in human milk from Bulgarian mothers; secondly, to reveal the core lactobacilli microbiota, using a culturable approach is not appropriate. Please clarify

Response 7: The aim of the current study was to find out the most common lactobacilli detected in Bulgarian samples of human breast milk and infant feces in the tested Bulgarian cohort from a relatively narrow region (Sofia region). Our study is the first one, because there are no other published data from Bulgaria in that research area. Our colleagues that we cited worked with only one strain of L. plantarum isolated from human breast milk and in their publication, there are no results for isolated other lactobacilli. There are no other data on the examination of more samples of breast milk in combination with infant’s feces from mother-infant tandem pairs. Therefore, we consider that our study is the first to present results for the most frequently isolated rod-shaped lactobacilli from such matrices originating from Bulgaria. Our further aim is not only to isolate but to investigate the newly isolated strains for their strain diversity and probiotic potential. We agree that the whole picture we will receive after NGS data and will be more correctly to use the word “core” than, that’s way we changed in the text with the most common lactobacilli or the most frequently detected, the most frequently found, etc.

REFERENCES

  1. Albesharat, R., Ehrmann, M. A., Korakli, M., Yazaji, S., and Vogel, R. F. (2011). Phenotypic and genotypic analyses of lactic acid bacteria in local fermented food, breast milk and faeces of mothers and their babies. Syst. Appl. Microbiol. 34, 148–155. doi: 10.1016/J.SYAPM.2010.12.001
  2. Martín, V., Maldonado-Barragán, A., Moles, L., Rodriguez-Baños, M., Campo, D., Fernández, L., et al. (2012). Sharing of bacterial strains between breast milk and infant feces. J. Hum. Lactat. 28, 36–44. doi: 10.1177/0890334411424729
  3. Soto, A. V., Martín, V., Jiménez, E., Mader, I., Rodríguez, J. M., and Fernández, (2014). Lactobacilli and Bifidobacteria in human breast milk: influence of antibiotherapy and other host and clinical factors. J. Pediatr. Gastroenterol. Nutr. 59, 78–88. doi: 10.1097/MPG.0000000000000347
  4. Ding, M., Qi, C., Yang, Z., Jiang, S., Bi, Y., Lai, J., et al. (2019). Geographical location specific composition of cultured microbiota and Lactobacillus occurrence in human breast milk in China. Food Funct. 10, 554–564. doi: 10.1039/C8FO02182A
  5. Solís, G., Reyes-Gavilan, C. G. D. L., Fernández, N.,Margolles, A., and Gueimonde, (2010). Establishment and development of lactic acid bacteria and bifidobacteria microbiota in breast-milk and the infant gut. Anaerobe 16, 307–310. doi: 10.1016/J.ANAEROBE.2010.02.004
  6. Jost, T., Lacroix, C., Braegger, C. P., Rochat, F., and Chassard, C. (2014). Vertical mother-neonate transfer of maternal gut bacteria via breastfeeding. Environ. 16, 2891–2904. doi: 10.1111/1462-2920.12238
  7. Murphy, K., Curley, D., O’Callaghan, T. F., O’Shea, C. A., Dempsey, E. M.,O’Toole, P. W., et al. (2017). The composition of human milk and infant faecal microbiota over the first three months of life: a pilot study. Sci. Rep. 7:40597. doi: 10.1038/srep40597
  8. Zhang X, Mushajiang S, Luo B,Tian F, Ni Y and Yan W (2020) The Composition and Concordance of Lactobacillus Populations of Infant Gut and the Corresponding Breast-Milk and Maternal Gut.Front. Microbiol. 11:597911. doi: 10.3389/fmicb.2020.597911
  9. Valérie Coeuret, Ségolène Dubernet, Marion Bernardeau, Micheline Gueguen, Jean Vernoux. Isolation, characterisation and identification of lactobacilli focusing mainly on cheeses and other dairy products. Le Lait, 2003, 83 (4), pp.269-306. ff10.1051/lait:2003019ff. ffhal-00895504f
  10. Hartemink R., Domenech V.R., Rombouts F.M., LAMVAB – A new selective medium for the isolation of lactobacilli from faeces, J. Microbiol. Meth. 29 (1997) 77–84
  11. Sakai, K. Oishi, T. Asahara, T. Takada, N. Yuki, K. Matsumoto, K. Nomoto, A. Kushiro M-RTLV agar, a novel selective medium to distinguish Lactobacillus casei and Lactobacillus paracasei from Lactobacillus rhamnosus Int. J. Food Microbiol., 139 (2010), pp. 154-160

Reviewer 2 Report

Comments and Suggestions for Authors

The authors have done a lot of microbiological work, but there are comments on the concept of the article and its presentation. 

Lines 141-147 - This information would be more appropriate in the results (to support the choice of method), but not in the introduction, as the authors do not compare the determination methodologies. 

The main comment on this manuscript is that there are no results comparing lactobacilli species from mother's milk and infant faeces. In the title, the authors make it clear that the milk microflora influences the formation of the faecal microflora of the child, but there is no comparison or parallels in the results. It seems that the authors finally set out to isolate as many different lactobacilli as possible and identified their species. 

The authors concluded in their conclusion about vertical transmission of lactobacilli, but I didn't see any results about this verticality. There are no charts or tables on the detection of the same strains or at least species in mother- infant pairs. 

Author needs to follow the concept of vertical transmission of lactobacilli and present the results in this, perhaps contribute tables, data analysis - statistics of detection of the same species in mother- infant pairs.

Author Response

Dear Reviewer,

We thank for the comments and recommendations made, which we will take into account.

We provide our response to the comments and remarks made.

Comment 1: Lines 141-147 - This information would be more appropriate in the results (to support the choice of method), but not in the introduction, as the authors do not compare the determination methodologies. 

Response 1: We agree with the reviewer's recommendation and have shortened this text in the Introduction section.

Comment 2 The main comment on this manuscript is that there are no results comparing lactobacilli species from mother's milk and infant faeces. In the title, the authors make it clear that the milk microflora influences the formation of the faecal microflora of the child, but there is no comparison or parallels in the results. It seems that the authors finally set out to isolate as many different lactobacilli as possible and identified their species. 

Response 2: We do not fully agree with this reviewer's remark, as the text of the manuscript draws a parallel between lactobacilli from breast milk and infant's feces based on the results obtained for their species diversity (lines 341-345 in the initial version of the manuscript and 376-385 in the revised version). The text has been edited to clarity, and a table has been presented to reflect it (Table 1). Our goal was not to isolate as many lactobacilli as possible, but to establish the culturable ones in the breast milk and feces of children with methodology chosen by us.

Comment 3: The authors concluded in their conclusion about vertical transmission of lactobacilli, but I didn't see any results about this verticality. There are no charts or tables on the detection of the same strains or at least species in mother- infant pairs. 

Response 3: In the main text of the manuscript, the detection of the same species in the breast milk samples and the corresponding sample of the children's feces is indicated (lines 341-345 in the initial version of the manuscript and 376-385 in the revised version) which supports the hypothesis of vertical transfer. In the Table 1 (in the revised version of the manuscript), we also shown that the species L. rhamnosus was simultaneously detected in the tandem samples of breast milk/feces (sample M – BM/F; sample KV – BM/F; sample N – BM/F); L. fermentum (sample EI – BM/ F; sample Y – BM/F); L. pararacei (sample KV – BM/F). Part of the data from the RAPD analysis also supports the hypothesis of the presence of a vertical transfer of lactobacilli from the mother to the child, although not all strains were analyzed with this method.

Comment 4: Author needs to follow the concept of vertical transmission of lactobacilli and present the results in this, perhaps contribute tables, data analysis - statistics of detection of the same species in mother- infant pairs.

Response 4: As we indicated above, the results supporting the hypothesis of vertical transfer of lactobacilli from the mother to the child are presented and discussed (lines 341-345 in the initial version of the manuscript and lines 376-385 and Table 1 in the revised version). Given the small number of analyzed samples, at this stage we have not presented statistics of the obtained results.

Reviewer 3 Report

Comments and Suggestions for Authors

The manuscript describes a study aimed at isolating and characterizing lactobacilli from breast milk in Bulgaria. It is reported that this is the first work carried out in this country and adds to all similar studies already carried out in other countries. Its regional importance is high. The study was aimed at identifying lactobacilli from the samples and, in fact, not lactic acid bacteria (LAB) since LAB is a broader term because it includes several other genera other than lactobacilli. Beyond this, which needs to be made clear throughout the text, the study carried out is well organized, with adequate analytical methodology and good presentation of results. The English is good but needs some writing corrections, which are detailed later. The text in general also needs revision since unnecessary repetitions are observed. The observations are detailed below:

- The title should inform that the study was aimed at isolating and identifying lactobacilli and not LAB. LAB includes other genera (Streptococcus, Enterococcus, etc.). For example, the abstract correctly refers to rod-shaped LAB but not the title and the text in general, including the Conclusions. Adapt the entire text to what was actually done

- The authors must justify why the central objective was only to characterize the presence of lactobacilli since one may wonder why not also investigate the presence of other genera of LAB

- Introduction: too long and heavy to read. It should be summarized a bit, avoiding, for example, describing the analytical methodology (lines 126-143) used since it should be detailed in Materials and Methods. In the Introduction, a brief reference can be made to the method used and its advantages. The comment on the origin of the human milk microbiota is also repeated (it appears in lines 97-115 and in lines 361-368). Mention it only in one place and in a more summarized way

-3.2: Fig. 1 is not cited in the text

Comments on the Quality of English Language

- In References, the genus and species of microorganisms should be written in italics. Review

- Idem lines 137, 309, 312-313, 324, 329, 345

- "lactobacilli", line 149, is not written in italics. It is not Latin, it is just a plural in English

- lines 297-8: they are plurals in English, do not use italics, not capital letters

Author Response

Dear Reviewer,

We thank for the comments, remarks and suggestions regarding the improvement of the manuscript, which will be taken into account. We provide our responses to the comments made on the manuscript.

Comment 1: The title should inform that the study was aimed at isolating and identifying lactobacilli and not LAB. LAB includes other genera (Streptococcus, Enterococcus, etc.). For example, the abstract correctly refers to rod-shaped LAB but not the title and the text in general, including the Conclusions. Adapt the entire text to what was actually done.

Response 1:  Yes, we know that LAB is a much broader term and includes a variety of species belonging to different genera, incl. and those that we have isolated. By using this term we did not mean to say that we have exhausted all genera of lactic acid bacteria. However, to the extent that we have not isolated species only from the genus Lactobacillus, but also other genera, such as Lactobacillus, Lacticaseibacillus, Limosilactobacillus, Lactiplantibacillus, Levilactobacillus (until recently belonging to the genus Lactobacillus), we have allowed ourselves to use the term lactic acid bacteria, since the specified genera really refer to LAB, without being the only ones. We will follow your recommendation and use the term lactobacilli to refer to genera with rod-shaped (bacillary) cells. We have made the necessary changes, both in the title and the main manuscript text.

Comment 2: The authors must justify why the central objective was only to characterize the presence of lactobacilli since one may wonder why not also investigate the presence of other genera of LAB

Response 2: The research presented by us is part of a project, the purpose of which at this stage is the establishment of lactobacilli by culture-dependent methods, which allows their isolation as pure cultures and their identification, for future research, in which their properties will be characterized, incl. potential probiotics. It is known that a number of species of lactobacilli have probiotic properties, which led us to study them as a priority. We have also isolated representatives of other genera of lactic acid bacteria, which will also be the subject of further research. We have justified our aim in the revised version of the manuscript (lines 169 – 182 in the revised version).

Comment 3: Introduction: too long and heavy to read. It should be summarized a bit, avoiding, for example, describing the analytical methodology (lines 126-143) used since it should be detailed in Materials and Methods. In the Introduction, a brief reference can be made to the method used and its advantages. The comment on the origin of the human milk microbiota is also repeated (it appears in lines 97-115 and in lines 361-368). Mention it only in one place and in a more summarized way

Response 3: We accept this remark and have complied with the recommendations made by revising the introduction, avoiding unnecessary methodological explanations.

Comment 4: Fig. 1 is not cited in the text

Response 4. A correction has been made and the Figure 1 (phylogenetic tree) is now cited in the text of the manuscript (line 461). In the revised version of the manuscript the former Figure 1 becomes Figure 2.  

Comment 5: In References, the genus and species of microorganisms should be written in italics. Review:- Idem lines 137, 309, 312-313, 324, 329, 345; "lactobacilli", line 149, is not written in italics. It is not Latin, it is just a plural in English; lines 297-8: they are plurals in English, do not use italics, not capital letters

Response 5: All the remarks you listed have been taken into account and the appropriate corrections have been made. All the corrections were highlighted in yellow.

Reviewer 4 Report

Comments and Suggestions for Authors

The pilot study by Asya Asenova et al. is of some interest to public health.

The text is well written, although the study design and outcomes are limited.

The manuscript needs to be majorly revised.

1. Please consider changing the manuscript type to "Brief Report", as you present the pilot study results. And revise the manuscript accordingly.

2. The gene name "16S rDNA" may be outdated. The terms "16S rDNA" and "16S rRNA" have been used interchangeably in some previous studies, however, I recommend avoiding writing "16S rDNA". Especially considering the fact that you used the term "16S rRNA" in M&M. Also, the American Society for Microbiology recommends using the "16S rRNA" term (https://doi.org/10.1128/cmr.17.4.840-862.2004).

3. Please, consider adding the figure summarising the results of taxonomical identification by mass-spectrometry and 16S rRNA sequencing to the main text of the manuscript.

4. If the authors have the data on the abundance of identified bacteria in studied samples (e.g. CFU/ml), I recommend adding this data to the manuscript and conducting statistical analysis.

5. Please, consider submitting the sequencing data to a public database and providing the accession number to the Data Availability Statement.

Author Response

Dear Reviewer,

Thank you for the review and the critical comments and suggestions for corrections.

We send our answer to your questions, suggestions and recommendations:

Comment 1: Please consider changing the manuscript type to "Brief Report", as you present the pilot study results. And revise the manuscript accordingly.

Response 1: Although we have stated that this is a pilot study, we believe that the experimental material and the obtained data are sufficient to be framed as an article. We made a lot of corrections to our manuscript and now generated results are much clearer. You can find all the corrections as highlighted in yellow.

Comment 2: The gene name "16S rDNA" may be outdated. The terms "16S rDNA" and "16S rRNA" have been used interchangeably in some previous studies, however, I recommend avoiding writing "16S rDNA". Especially considering the fact that you used the term "16S rRNA" in M&M. Also, the American Society for Microbiology recommends using the "16S rRNA" term (https://doi.org/10.1128/cmr.17.4.840-862.2004).

Response 2: We accept your recommendation and the term “16S rDNA was replaced by “16S rRNA” throughout the manuscript.

Comment 3: Please, consider adding the figure summarizing the results of taxonomical identification by mass-spectrometry and 16S rRNA sequencing to the main text of the manuscript.

Response 3: We understand the reviewer's recommendation, but because the table is too large, we decided to keep it as a supplementary table (Supplementary Table S7).

Comment 4: If the authors have the data on the abundance of identified bacteria in studied samples (e.g. CFU/ml), I recommend adding this data to the manuscript and conducting statistical analysis.

Response 4: We have no data on the abundance of identified species, as their isolation was done after an enrichment procedure. As an additional information that we could provide was that in parallel we seeded breast milk samples on Columbia agar with 5% sheep blood and revealed that there were about 103-105 cfu/ml.

Comment 5: Please, consider submitting the sequencing data to a public database and providing the accession number to the Data Availability Statement.

Response 5: You may have missed seeing that the sequences were submitted to NCBI and their numbers are cited on lines 232 and 233 in the Materials and Methods section (lines 232-233 in the initial version of the manuscript and 260-261 in the revised version). After the revision we noticed them in the Table S7 also. However, we have included the access sites to the Materials and Methods of our manuscript.

Round 2

Reviewer 1 Report

Comments and Suggestions for Authors

I find the author's response to be highly satisfactory. The manuscript can be accepted in its current form.

Author Response

Dear Reviewer,

Thank you very much for your valuable comments to our manuscript. We highly appreciate your efforts in providing us a such detail comments. 

Kind Regards,

Iliyana Rasheva 

Reviewer 2 Report

Comments and Suggestions for Authors

I think these sentences are better included in the conclusion: line 177-182

I should note the article has been improved, however, the visualisation of the results has remained low. The article is not interesting for a wide range of readers. 

I remain of the opinion that Table 1 is very uninformative.

Please add the practical implications and scientific perspectives of the study.

Title - aim - conclusions , these should be in continuity. The authors should check this flow. Currently, I see some inconsistency here. 

Author Response

Comment 1:  I think these sentences are better included in the conclusion: line 177-182

Response 1: We agree with the reviewer's recommendation and removed and partially revised it in the Conclusion section.

Comment 2: “the visualisation of the results has remained low”

Response 2: We do not fully agree with the reviewer’s comment concerning the visualization of the results. Most of the data obtained in our manuscript relate to strain isolation and identification. The current microbiological analyses generated data that cannot be visualized by sophisticated graphs or tables. We demonstrated our results in the main text with 3 figures (species distribution, phylogenetic tree and dendrogram clustering of L. rhamnosus and L. paracasei isolates according to their RAPD profiling), as this helps better understanding and summarizes the data. In the Supplementary materials we have sufficient additional information presented in tables.    

Comment 3: “The article is not interesting for a wide range of readers”.

Response 3: We do not agree with the reviewer’s comment. In the recent years, there has been a growing interest in studying the human milk microbiome as well infant feces microbiome using more sophisticated culture-dependent and culture-independent techniques, as well as ‘omics approaches’. For example Dombrowska-Pali et al., 2024, published a review of the human milk microbiome and they included a table “Diversity of the human milk microbiome depending on geographical location”. In this table they were described data from European countries like Finland, Spain, Italy, Slovenia, and Switzerland. Unfortunately, it does not contain data from Bulgaria or other countries from the Balkan Peninsula. One of the potential reason probably was that published data from this geographic area are very limited. We believe that with our publication this gap will be filled at least partially and in future publications worldwide our data could be included. Our report on the most common isolated lactobacilli will enhance the knowledge about the cultivable lactobacilli originating from human breast milk and infant feces based on culturomics techniques. Also, considering the lack of previously published data from our region, makes our findings even more notable. Even though our study contains information related to only lactobacilli in breastmilk and neonate stool, it will stand as a base study for Bulgaria, and thereafter may prompt scientific interest in the field of breastmilk and infant fecal microbiota. We really believe that articles in the field of microbiome will generate a lot of readers, viewers, citations, downloads etc. At this stage as a preprint, our manuscript has 52 views and 89 downloads (https://www.preprints.org/manuscript/202408.0196/v1).

On the other hand, each research area has a limited readers. The interest to human microbiome at all has gained enormous interest during recent years, not only from the scientific audience but attract more and more readers from the society. This expand the influence of the science at all. The science findings about the microbiome of human breast milk and its relation with infant feces microbiome, have a paramount social impact role concerning the significant of breastfeeding and could also motivate the volunteer mothers to donate their milk. There are many publication in the field and they have a very high rang.

Reference

  1. Dombrowska-Pali, A.; Wiktorczyk-Kapischke, N.; Chrustek, A.; Olszewska-SÅ‚onina, D.; Gospodarek-Komkowska, E.; Socha, M.W. Human Milk Microbiome—A Review of Scientific Reports. Nutrients 2024, 16, 1420. https://doi.org/10.3390/ nu16101420

Comment 4: “I remain of the opinion that Table 1 is very uninformative”.

Response 4: We suppose that the main idea of this table remained unclear for the reviewer. Table 1 in the main text of the current manuscript include information for the distribution of the newly isolated lactobacilli from one species within one tandem mother-infant pair. We think it is informative because demonstrate the potential of vertical bacteria transfer as well the relation between mother and infant gut microbiome. In fine, we made some changes in the table.

Comment 5: “Please add the practical implications and scientific perspectives of the study”.

Response 5: Thank you very much for this remark. We added some comments concerning the practical implications and scientific perspectives of the study. All changes are marked in green in the revised manuscript.

Culturomics techniques that we conducted in our research (culture methods in in combination with MALDI TOF MS and 16S rRNA sequencing) can enhance the ability to analyse the human milk microbiota in vitro and providing a more detailed understanding of the microbial communities in this complex environment. By maximizing the isolation of diverse lactobacilli, we can gain insight into diversity of the human milk microbiome depending on geographical locationits. The knowledge of typical microbiota can lead to developing of strategies targeting breastmilk microbiota to impact both maternal and infant heath. Also the future scientific perspectives of the current study include experiments concerning the probiotic potential of the isolated strains. Many similar publications from different countries can be found in various scientific journals. For example Cheema, S et al., 2023, Wang, F., et al, 2024.

References

  1. Cheema, S.; Li, R.; and Cameron, S.; Culturomics as a tool to better understand the human milk microbiota and host–microbiota interactions.Microbiota and Host. 2023, no. 1, doi:https;//doi.org/10.1530/MAH-23-0001.
  2. Wang, F.; Yu, L.; Ren, Y.; Zhang, Q.; He, S.; Zhao, M.; He, Z.; Gao, Q.; and Chen, J.; 2024,An optimized culturomics strategy for isolation of human milk microbiota. Microbiol. 15:1272062. doi: 10.3389/fmicb.2024.1272062.

Comment 6: “Title - aim - conclusions , these should be in continuity. The authors should check this flow. Currently, I see some inconsistency here”. 

Response 6: We revised the aim and conclusion according the reviewer recommendations. All changes are marked in green in the revised manuscript.

Reviewer 4 Report

Comments and Suggestions for Authors

The authors improved the quality of the report.

However, I still think that the manuscript type "Brief Report" is more suitable for this pilot study, as the reported data is limited.

I would like to leave the decision of this question to the Editor's consideration.

Author Response

Dear Reviewer,

Thank you very much for your valuable comments to our manuscript. We highly appreciate your efforts in providing us your comments.

Kind Regards,

Iliyana Rasheva